# Couple’s Relationship and Depressive Symptoms during the Transition to Parenthood and Toddler’s Emotional and Behavioral Problems

**DOI:** 10.3390/ijerph19063610

**Published:** 2022-03-18

**Authors:** Tiago Miguel Pinto, Bárbara Figueiredo

**Affiliations:** School of Psychology, University of Minho, 4710-057 Braga, Portugal; bbfi@psi.uminho.pt

**Keywords:** couple’s interaction, mother’s and father’s depressive symptoms, transition to parenthood, toddler’s internalizing and externalizing problems

## Abstract

The couple’s relationship and mother and father’s depressive symptoms during the transition to parenthood were associated with the toddler’s emotional and behavioral problems. This study aimed to analyze how the couple’s positive and negative interactions and mother and father’s depressive symptoms during the transition to parenthood impact toddlers’ emotional and behavioral problems. A sample of 95 mothers and fathers (*N* = 190) were recruited and individually completed questionnaires to assess couples’ positive and negative interactions and depressive symptoms during the first trimester of pregnancy and at 3 and 30 months postpartum, and they completed the Child Behavior Checklist 1.5–5 at 30 months postpartum. The path analyses revealed that the couple’s postnatal negative interaction partially mediates the impact of the mother’s prenatal depressive symptoms on the toddler’s internalizing problems at 30 months postpartum. The father’s postnatal depressive symptoms and the couple’s concurrent positive interaction mediated the impact of the couple’s prenatal positive interaction on the toddler’s externalizing problems at 30 months postpartum. The screening of the couple’s negative interaction and depressive symptoms during pregnancy and the postnatal period can help to identify mothers, fathers, and toddlers at risk for mental health problems.

## 1. Introduction

The transition to parenthood is a major life event that leads both women and men to perform several developmental tasks to adapt to several biological, psychological, and sociocultural changes. The positive performance of these developmental tasks allows an adaptive transition to parenthood, leading to psychological adjustment and development and preparing for adequate parenting. In contrast, a negative performance of these developmental tasks may lead to an increase of psychopathological symptoms and inadequate parenting [1,2]. It comprises changes both in the individual and in the family subsystems (e.g., marital, parental, and coparental) [1]. According to family systems theory, the arrival of a new child demands a reorganization of the family structure, encompassing a network of interactions and dynamics among mothers and fathers’ physical, cultural, and social backgrounds [3,4,5]. Thus, mother and father’s psychological adjustment to the transition to parenthood could be key elements for child development and mental health [6,7,8]. Internalizing and externalizing problems are symptoms of child mental health problems. Internalizing problems include anxiety/depression symptoms, somatic complaints, withdrawal, and sleep problems, and externalizing problems include attention problems and aggressive behavior [9].

The links between the couple’s relationship and the toddler’s emotional and behavioral problems are supported by longitudinal research. The couple’s relationship includes positive and negative interactions. Positive interactions include a sense of support and care, affection, closeness, and joint interests and activities shared by the couple, while negative interactions include irritability, arguments, and criticisms between the couple [10]. Studies reported the negative impact of the couple’s negative interaction (e.g., conflict) on the toddler’s internalizing and externalizing problems [11,12]. Contrarily, the couple’s positive interaction was associated with higher emotional and behavioral adjustment in toddlers [13,14]. However, literature on the impact of a couple’s interaction during the transition to parenthood on the toddler’s emotional and behavioral adjustment is still scarce.

Literature provides evidence on the adverse impact of mothers’ or/and fathers’ depressive symptoms during the transition to parenthood on toddlers’ emotional and behavioral problems. Although mostly in different studies, the adverse impact of both mothers’ prenatal and postnatal depressive symptoms on toddlers’ internalizing and externalizing problems have been reported [7,15,16]. Fathers’ postnatal depressive symptoms were also found to negatively impact toddlers’ internalizing and externalizing problems [7,16]. Moreover, after controlling for the mother’s postnatal depressive symptoms, the father’s postnatal depressive symptoms were found to double the risk of toddler’s emotional and behavioral problems [7]. A metanalytic analysis [17] reported that the interdependency between spouses seems to be reflected in depression comorbidity among parents, allowing speculation that the father may be at risk when the mother is experiencing depression during the transition to parenthood, and vice-versa.

The literature has been exploring how the couple’s relationship and mothers’ and fathers’ depressive symptoms can interact and impact children’s internalizing and externalizing problems. Namely, parental depressive symptoms were found to mediate the impact of a couple’s relationship on children’s internalizing and externalizing problems [18,19]. Thus, exploring how the couple’s relationship and mothers’ and fathers’ depressive symptoms during the transition to parenthood could interact and impact toddlers’ emotional and behavioral development could be a major contribute contribution to the literature on family and developmental psychopathology. This study aimed to clarify how the couple’s positive and negative interactions and depressive symptoms during the transition to parenthood impact the toddler’s emotional and behavioral problems. Specifically, a model was designed to test the following hypothesis: (1) the impact of prenatal couple’s positive interaction on toddler’s externalizing problems at 30 months could be mediated by fathers’ postnatal depressive symptoms postpartum and/or by the concurrent couple’s positive interaction, and (2) the impact of mothers’ prenatal depressive symptoms on toddler’s internalizing problems at 30 months could be mediated by the postnatal couple’s negative interaction.

## 2. Materials and Methods

### 2.1. Participants and Procedures

The sample comprised 95 mothers and fathers (*N* = 190) from a larger longitudinal study aiming to analyze mothers’ and fathers’ perinatal mental health and child development. The present research was conducted in accordance with the Helsinki Declaration and received previous approval from the Ethical Commission of all institutions involved. Couples were recruited at a public Health Service in Northern Portugal during the first trimester of pregnancy. The exclusion criteria were not reading or writing Portuguese and having multiple gestations, considering the reported association between twin parenthood and increased parents’ mental health problems [20]. The aims and the procedures of the study were explained, and the parents willing to participate signed a written consent form. This study had a longitudinal design with three assessment waves: first trimester of pregnancy (8–14 gestational weeks, *M* = 12.97, *SD* = 1.49) and 3 (10–14 postpartum weeks, *M* = 13.64, *SD* = 0.81) and 30 months postpartum (24–36 months postpartum, *M* = 38.78, *SD* = 1.55). A socio-demographic questionnaire and measures of couple’s positive and negative interactions and depressive symptoms were independently completed by mothers and fathers at each assessment wave. Mothers and fathers completed the Children Behavior Checklist (CBCL) 1.5–5 at 30 months postpartum. The assessment waves were selected (1) to assess couple’s interaction and depressive symptoms during the beginning and the end of the transition to parenthood, (2) to maintain a time frame of 3 months between each assessment wave [10], and (3) to assess toddlers early in the preschool period.

From the 260 mothers and fathers (*N* = 520) that completed the first assessment wave, 128 mothers and 116 fathers (*n* = 244; 49.9%) completed the CBCL at 30 months postpartum. Parents who completed all assessment waves were different from those who did not complete all assessment waves, regarding their socioeconomic level, age, occupational status, and depressive symptoms at the first trimester of pregnancy. They were more likely to belong to a medium/high socio-economic level, χ^2^(2) = 26.45, *p* < 0.001, be older, χ^2^(3) = 33.05, *p* < 0.001, be employed, χ^2^(6) = 16.32, *p* = 0.012, and report less depressive symptoms at the first trimester of pregnancy, *t*(503) = 4.05, *p* < 0.001. CBCL data were included only when all the study measures were completed by both parents (*N* = 190; *n* = 95 mothers; *n* = 95 fathers). The mothers and fathers who completed the CBCL did not differ from those who did not complete the data on mothers’ and fathers’ sociodemographic characteristics, couple’s positive and negative interactions, and depressive symptoms during the first trimester of pregnancy, at 3 and 30 months postpartum. In addition, no differences were found between the completed and uncompleted data on toddlers’ sociodemographic and biometric data and CBCL scores at 30 months postpartum.

### 2.2. Measures

#### 2.2.1. Socio-Demographic Measures

A self-administered questionnaire was used to collect mothers, fathers, and toddlers’ socio-demographic information.

#### 2.2.2. Couple’s Positive and Negative Interactions

The Relationship Questionnaire (RQ) [21] was used to assess the couple’s positive and negative interactions. It is a 12-item self-report questionnaire, rated on a four-point Likert-type scale (1–4), and comprises two subscales: positive interaction (eight items) and negative interaction (four items). Higher scores on these subscales indicate more positive or more negative interactions. The RQ has shown good psychometric characteristics [21]. In the present study, Cronbach’s alphas ranged from 0.70 to 0.92.

#### 2.2.3. Depressive Symptoms

The Edinburgh Postnatal Depression Scale (EPDS) [22] was used to assess mothers’ and fathers’ depressive symptoms. The EPDS is a 10-item self-report questionnaire scored on a four-point Likert-type scale, designed to assess the intensity of depressive symptoms within the previous seven days. This instrument has been used in several studies with mothers and fathers during pregnancy and the postpartum period [10,23]. EPDS Portuguese version showed good internal consistency in mothers and fathers [10,24]. In the present study, Cronbach’s alpha ranged from 0.78 to 0.88.

#### 2.2.4. Emotional and Behavioral Problems

The Child Behavior Checklist 1.5–5 [9] was used to assess toddlers’ internalizing and externalizing problems. This instrument has 99 items that describe emotional and behavioral problems in toddlers of 1.5 to 5 years of age. Parents are requested to rate their toddler’s behavior during the last 2 months on a three-point Likert-type scale. The instrument assesses eight behavioral problems, and seven among them are grouped in two subscales: internalizing behavior (emotional reactivity, anxious/depressed, somatic complaints, and withdrawal) and externalizing behavior (attention problems and aggressive behavior). The Portuguese version of the instrument has shown good internal consistency and validity [25]. In the present sample, Cronbach’s alpha coefficient was 0.90 for both internalizing and externalizing problems.

### 2.3. Statistical Analyses

A composite score was computed for the couple’s positive and negative interactions at each assessment wave and for toddlers’ internalizing and externalizing scores (average score of the mothers’ and the fathers’ positive and negative interaction scores and internalizing and externalizing scores). Preliminary analyses were performed to identify potential covariates of the independent variables (IV; mothers’ depressive symptoms and couple’s positive interaction at the first trimester of pregnancy), hypothesized mediators (couple’s negative interaction and fathers’ depressive symptoms at 3 months postpartum and couple’s positive interaction at 30 months postpartum), and the dependent variables (DV; toddler’s internalizing and externalizing symptoms at 30 months postpartum). Multivariate analyses of variance (MANOVAs) were performed with the independent and with the hypothesized mediators according to parental socio-demographic variables (see Section 3.1). Likewise, univariate analyses of variance (ANOVAs) were performed with the dependent variables according to the toddler’s socio-demographic and biometric variables (see Section 3.1).

A longitudinal path analysis model was tested using structural equation modeling (SEM) to analyze how the couple’s positive and negative interactions and depressive symptoms during the transition to parenthood impact the toddler’s emotional and behavioral problems. Considering the study’s hypothesis, the model included the couple’s positive interaction and mothers’ depression scores at the first trimester of pregnancy as IVs (exogenous). The couple’s positive interaction at the first trimester of pregnancy was directly linked with four possible parameters: (1) the couple’s negative interaction at 3 months postpartum, (2) the couple’s positive interaction at 30 months postpartum, (3) the father’s depressive symptoms at 3 months postpartum (hypothesized mediators), and (4) the toddler’s internalizing symptoms at 30 months postpartum (DV; endogenous). Mothers’ depressive symptoms at the first trimester of pregnancy were directly linked with five possible parameters: (1) the couple’s negative interaction at 3 months postpartum, (2) the couple’s positive interaction at 30 months postpartum, (3) the father’s depressive symptoms at 3 months postpartum (hypothesized mediators); and (4) the toddler’s internalizing and (5) externalizing scores at 30 months postpartum (DVs; endogenous). The couple’s negative interaction and fathers’ depressive symptoms at 3 months postpartum were directly linked with the couple’s positive interaction at 30 months postpartum and the toddler’s internalizing and externalizing symptoms at 30 months postpartum. The couple’s positive interaction at 30 months postpartum was directly linked with the toddler’s externalizing symptoms at 30 months postpartum. Variables assessed at the same assessment wave were correlated (see Figure 1). The model was constructed with observed variables, and maximum likelihood estimation was applied. The initial model was submitted to the Wald test modification indices to increase the goodness of fit indices. Following recommendations [26], indices from different classes were used to assess model goodness of fit. Mediation effects were identified through decomposition of effects results and tested following guidelines [27]. Analyses were performed using the SPSS and AMOS SPSS 24 (SPSS Inc., IBM, Armonk, NY, USA).

## 3. Results

### 3.1. Participant’s Sociodemographic Characteristics

Parents were married or cohabiting. Most parents were Portuguese (94.7% of mothers and 91.5% of fathers), were from the medium to high socioeconomic levels (84.1% of mothers and 84.9% of fathers), had nine or more years of education (84.2% of mothers and 72.3% of fathers), and were employed (86.3% of mothers and 91.5% of fathers). Mothers’ ages ranged between 20 and 42 years (*M* = 30.29, *SD* = 4.95) and fathers’ ages ranged between 18 and 46 (*M* = 32.80, *SD* = 5.52). More than half were first-time parents (64.2%). 

The toddlers were 32-months old (*M* = 32.06, *SD* = 6.50), and they had no health problems reported by mothers or fathers. Most toddlers were born at term (≥37 gestational weeks; 93.7%), 57.9% were male, and 49.5% were attending preschool. No differences (all *ps* > 0.05) were found on the studied variables according to parental and toddler socio-demographic variables.

### 3.2. Path Analyses Model: The Impact of Couple’s Positive and Negative Interactions and Depressive Symptoms during Pregnancy and the Postpartum Period on Toddler’s Emotional and Behavioral Problems

#### 3.2.1. Initial Model

The initial model presented acceptable-to-bad fit indices, χ^2^(2) = 5.86, IFI = 0.98, CFI = 0.98, SRMR = 0.03, and RMSEA = 0.14 (see Figure 1a). The Wald test results indicated that six parameters needed to be removed from the initial model to achieve a good fit (see Table 1).

#### 3.2.2. Final Model

After removing the six parameters (see Table 1), the final model presented good fit indices, χ^2^(8) = 12.00, CFI = 0.98, IFI = 0.98, SRMR = 0.06, RMSEA = 0.07. The chi-squared test for nested models’ comparison indicated that the fit of the model has not been significantly hindered by introducing the additional constraints suggested by the Wald test, χ^2^(6) = 6.14 and *p* = 0.41. Therefore, the increase in the chi-squared is not significant in reducing the model fit, and the other model goodness of fit indices were better for the final model. The analyses of the model paths revealed that the higher couple’s positive interaction at the first trimester of pregnancy was associated with the lower couple’s negative interaction and fewer fathers’ depressive symptoms at 3 months postpartum. Higher couple’s positive interaction at the first trimester of pregnancy was also associated with higher couple’s positive interaction at 30 months postpartum. More mothers’ depressive symptoms at the first trimester of pregnancy were associated with higher couple’s negative interaction and more fathers’ depressive symptoms at 3 months postpartum and more toddler’s internalizing and externalizing problems at 30 months postpartum. Higher couple’s negative interaction and more father’s depressive symptoms at 3 months postpartum were associated with more toddler’s internalizing problems at 30 months postpartum. Higher couple’s positive interaction at 30 months postpartum was associated with fewer toddler externalizing problems at 30 months postpartum (see Figure 1b).

#### 3.2.3. Mediation Analysis

The decomposition of the associations in the final model noted two significant indirect associations, suggesting possible mediation links. The couple’s negative interaction at 3 months postpartum was tested as a meditator on the impact of the mother’s depressive symptoms at the first trimester of pregnancy on the toddler’s internalizing problems at 30 months postpartum. The mediation paths a (*B* = 0.03, *SE* = 0.01), b (*B* = 0.21, *SE* = 0.05), c (*B* = 0.03, *SE* = 0.01), and c’ (*B* = 0.03, *SE* = 0.08) were all significant (all *ps* < 0.05), indicating the presence of both direct and indirect associations. The couple’s negative interaction at 3 months postpartum partially mediated the impact of the mother’s depressive symptoms at the first trimester of pregnancy on the toddler’s internalizing problems at 30 months postpartum.

Father’s depressive symptoms at 3 months postpartum and couple’s positive interaction at 30 months postpartum were tested as mediators of the impact of couple’s positive interaction at the first trimester of pregnancy on toddler’s externalizing problems at 30 months postpartum. Results indicated the presence of significant indirect links of the father’s depressive symptoms at 3 months postpartum (*B* = −1.77, *SE* = 0.82) and the couple’s positive interaction at 30 months postpartum (*B* = −2.29, *SE* = 1.27) on the toddler’s externalizing problems at 30 months postpartum. Paths a and b were significant both in the mediation model of fathers’ depressive symptoms at 3 months postpartum (path a, *B* = −0.44, *SE* = 0.13; path b, *B* = 4.80, SE = 1.41) and in the mediation model of the couple’s positive interaction at 30 months postpartum (path a, *B* = 0.28, *SE* = 0.04; path b, *B* = −11.00, *SE* = 3.61). Path c was also significant when the mediating paths (a and b) for the two mediators were fixed at zero (*B* = −4.90, *SE* = 1.85). Path c’ was non-significant both in the mediation model of fathers’ depressive symptoms at 3 months postpartum (*B* = −3.37, *SE* = 1.90) and in the mediation model of couple’s positive interaction at 30 months postpartum (*B* = −2.60, *SE* = 2.20). Results indicate a double mediation, as the Wald test suggests a better model fit without having a path c’ parameter. Both father’s depressive symptoms at 3 months postpartum and the couple’s positive interaction at 30 months postpartum mediated the impact of the couple’s positive interaction at the first trimester of pregnancy on the toddler’s externalizing problems at 30 months postpartum (see Figure 2).

## 4. Discussion

This study tested an explanatory model to analyze how the couple’s positive and negative interactions and mother’s and father’s depressive symptoms during the transition to parenthood impact the toddler’s emotional and behavioral problems. Findings provided evidence on the negative impact of the mother’s prenatal depressive symptoms on the toddler’s internalizing and externalizing problems at 30 months. This result is in line with previous studies reporting the adverse impact of mothers’ prenatal depressive symptoms on toddlers’ internalizing and externalizing problems [15,16,28]. This adverse impact could be explained by the negative influence of depressive symptoms on the utero environment, which can alter fetal development and increase the susceptibility for later emotional and behavioral problems [23,28,29]. 

Findings provided evidence on the possible mediator role of the couple’s postnatal negative interaction in the impact of mother’s prenatal depressive symptoms on toddler’s internalizing problems at 30 months. This result is congruent with previous data suggesting the impact of the mother’s depressive symptoms on the couple’s relationship quality and satisfaction during the transition to parenthood [30]. Besides the adverse impact on fetal development, the mother’s prenatal depressive symptoms could also increase the negative interactions between the couple during the postpartum period that could increase the toddler’s internalizing problems. 

Findings provided evidence regarding the mediator role of the father’s postnatal depressive symptoms in the impact of the couple’s prenatal positive interaction on toddler’s externalizing problems at 30 months postpartum. This result is in line with previous studies suggesting the protective role of couple’s positive interaction on father’s depressive symptoms during the transition to parenthood [10] and showing the adverse impact of father’s postnatal depressive symptoms on toddler’s externalizing problems [7]. The results of the present study integrate and advance previous findings by suggesting that more couples’ positive interactions early in pregnancy decrease the father’s postnatal symptoms, which in turn decreases the toddler’s externalizing problems. Fathers with fewer postnatal depressive symptoms may have a higher ability to interact with the infant, presenting a higher sensibility to adequately respond to the infant’s signals. The father’s sensitivity during interactions with the infant was found to predict better behavioral and psychological development, while disengaged interactions between fathers and their infants were found to predict externalizing behavioral problems in toddlers [31,32,33].

Findings also provided evidence that the couple’s concurrent positive interaction mediated the impact of the couple’s prenatal positive interaction on the toddler’s externalizing problems at 30 months postpartum. The couple’s relationship has been extensively associated with the toddler’s externalizing and internalizing problems [8,18,34]. Findings from the present study suggested that the couple’s prenatal positive interaction is associated with the later concurrent couple’s positive interaction, which, in turn, decreases the toddler’s externalizing problems. This result supports the importance of the concurrent couple’s positive interaction in protecting the toddler from externalizing problems, as previously suggested [35]. Although the transition to parenthood is marked by a decrease in the couple’s positive interaction, couples who maintain more positive interactions during the transition to parenthood were found to present better psychological adjustment during the postpartum period [10,36,37].

### 4.1. Strengths and Limitations

Strengths of the present study include using a longitudinal design to test a complex model to analyze how the couple’s positive and negative interactions and mother’s and father’s depressive symptoms during the transition to parenthood impact the toddler’s emotional and behavioral problems. Some limitations should also be considered. A higher sample size could allow testing a model with higher complexity. The differences found between the participants that completed and did not complete all the assessment waves could compromise the generalization of findings. Additional limitations include the use of self-report measures and the impossibility of testing other possible predictors.

### 4.2. Implications for Clinical Practice and Research

This study has major implications for clinical practice and research. The screening of the couple’s negative interaction and depressive symptoms during pregnancy and the postnatal period can help to identify mothers, fathers, and toddlers at risk for mental health problems. Promotion and prevention programs during the transition to parenthood should start by decreasing the couple’s negative interaction and parents’ depressive symptoms during pregnancy. This could be the first step to promote an adjusted transition to parenthood and subsequently to promote mother, father, and toddler’s mental health. 

Future studies could test more complex models to enlarge the comprehension of how the couple’s relationship and mother and father’s depressive symptoms impact the toddler’s emotional and behavioral problems. Namely, it could be relevant to explore other mediators (e.g., fetal susceptibility markers) and moderators (e.g., toddler’s sex) on the impact of couple’s interaction and mothers’ and fathers’ depressive symptoms on toddler’s emotional and behavioral problems.

## 5. Conclusions

This study provides a contribution to the literature on family and developmental psychopathology by proposing pathways for how the couple’s positive and negative interactions and mother’s and father’s depressive symptoms during the transition to parenthood impact the toddler’s emotional and behavioral problems.

## Figures and Tables

**Figure 1 ijerph-19-03610-f001:**
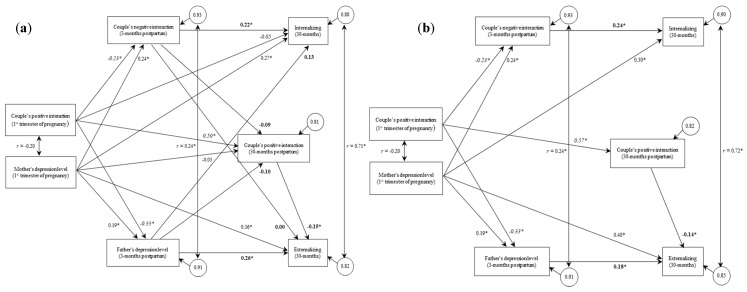
Initial (**a**) and final model (**b**) with standardized values. * *p* < 0.05.

**Figure 2 ijerph-19-03610-f002:**
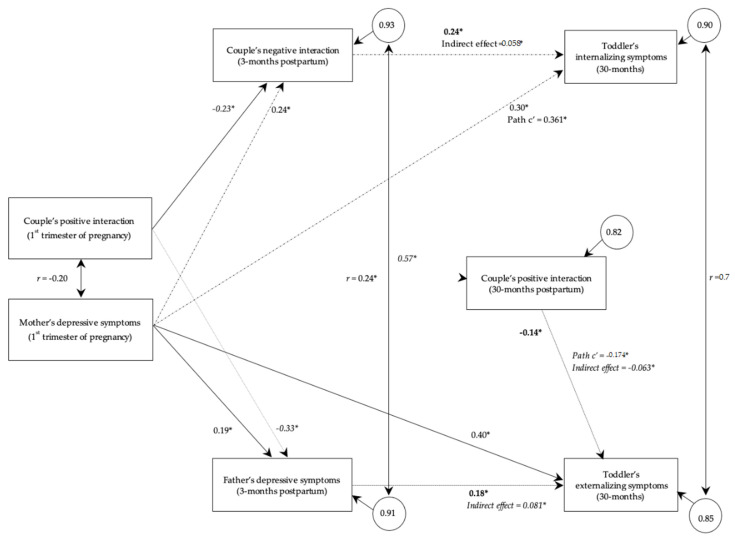
Mediation analyses with standardized values for all parameters. * *p* < 0.05.

**Table 1 ijerph-19-03610-t001:** Multivariate Wald test results of the initial model.

Steps	Parameters Suggested for Removal	Cumulative Increment	
		χ^2^(df)	*p*
1	Couple’s negative interaction at 3 months postpartum– Toddler’s externalizing problems at 30 months postpartum	0.001(1)	0.97
	Mothers’ depression symptoms at the first trimester of pregnancy– Couple’s positive interaction at 30 months postpartum	0.094(2)	0.95
2	Couple’s positive interaction at the first trimester of pregnancy– Toddler’s internalizing problems at 30 months postpartum 2	0.471(1)	0.49
	Couple’s negative interaction at 3 months postpartum– Couple’s positive interaction at 30 months postpartum	1.676(2)	0.43
	Fathers’ depression symptoms at 3 months postpartum– Couple’s positive interaction at 30 months postpartum	3.920(3)	0.27
	Fathers’ depression symptoms at 3 months postpartum– Toddler’s internalizing problems at 30 months postpartum	6.175(4)	0.19
3	Couple’s positive interaction at the first trimester of pregnancy– Mothers’ depression symptoms at the first trimester of pregnancy 3	3.722(1)	0.04

Note. The parameter from Step 3 was not withdrawn as *p* < 0.05 was achieved.

## Data Availability

The data presented in this study are available on request from the corresponding author. The data are not publicly available due to ethical issues.

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
