# Peer review of "Couple’s Relationship and Depressive Symptoms during the Transition to Parenthood and Toddler’s Emotional and Behavioral Problems"

_ijerph, 2022, doi:10.3390/ijerph19063610_

Round 1
Reviewer 1 Report
An important article with interesting findings. Please see my comments below:
Couples relationship and depressive symptoms during the transition to parenthood and toddlers emotional and behavioral problems
- Try to simplify the language used to describe the relationship between the different variables for a non-expert reader. For example, could “mood changes” be used in place of “depressive symptoms” and “fathers and mothers” to “parents?”
- Similarly, try to simplify the study objective as, at the moment, it compels to read more than once to make sense of what it means. Interaction at X mediates Y, which leads to Z. I am not suggesting overhauling the introduction and methods, but a more simplified text would help.
- For the statements below: Negative interaction at 3 months postpartum partially mediates the mother’s depressive symptoms at the first trimester…the first trimester of the following pregnancy or is this retrospective referring to the current baby? Please specify: current pregnancy or subsequent?
The path analyses revealed that couples’ negative interaction at 3 months postpartum partially mediates the impact of mother’s depressive symptoms at the 1st trimester of pregnancy on toddler’s internalizing problems at 30 months post-partum. Father’s depressive symptoms at 3 months postpartum and couples’ positive interaction at 30 months postpartum mediated the impact of couples’ positive interaction at the 1st trimester of pregnancy on toddler’s externalizing problems at 30 months postpartum.
- How was filling of the CBCL by parents controlled?
- The manuscript will benefit from an extensive English revision by a native speaker.
Thanks.
Author Response
Reviewer’s comment 1
Comments to the Author
An important article with interesting findings. Please see my comments below:
Couples relationship and depressive symptoms during the transition to parenthood and toddlers emotional and behavioral problems
Response to reviewer’s comment 1
Thank you so much for your suggestions!
Reviewer’s comment 2
Try to simplify the language used to describe the relationship between the different variables for a non-expert reader. For example, could “mood changes” be used in place of “depressive symptoms” and “fathers and mothers” to “parents?”
Response reviewer’s comment 2
As you suggested, we have tried to simplify the language through the manuscript.
Reviewer’s comment 3
Similarly, try to simplify the study objective as, at the moment, it compels to read more than once to make sense of what it means. Interaction at X mediates Y, which leads to Z. I am not suggesting overhauling the introduction and methods, but a more simplified text would help.
Response to reviewer’s comment 3
As you suggested, we have tried to simplify the study objective.
Reviewer’s comment 4
For the statements below: Negative interaction at 3 months postpartum partially mediates the mother’s depressive symptoms at the first trimester…the first trimester of the following pregnancy or is this retrospective referring to the current baby? Please specify: current pregnancy or subsequent?
Response to reviewer’s comment 4
We revised the sentence in order to make clear that it is about the current pregnancy.
Reviewer’s comment 5
The path analyses revealed that couples’ negative interaction at 3 months postpartum partially mediates the impact of mother’s depressive symptoms at the 1st trimester of pregnancy on toddler’s internalizing problems at 30 months post-partum. Father’s depressive symptoms at 3 months postpartum and couples’ positive interaction at 30 months postpartum mediated the impact of couples’ positive interaction at the 1st trimester of pregnancy on toddler’s externalizing problems at 30 months postpartum.
How was filling of the CBCL by parents controlled?
Response to reviewer’s comment 5
The parents were instructed to independently complete the CBCL (please see page 3).
Reviewer’s comment 6
The manuscript will benefit from an extensive English revision by a native speaker.
Response to reviewer’s comment 6
The English were revised throughout the manuscript.
Reviewer 2 Report
Please find my comments below:
This is an interesting paper that seeks to elucidate the way interaction among the parents and their depressive symptoms during the transition to parenthood predict emotional and behavioral problem in toddlers. The research work is sound and findings are well presented. However, I have few minor comments regarding the manuscript.
- Kindly recheck the manuscript for language and grammatical errors. To enumerate a few
Line 21: Screening couples’ interaction and depressive symptoms early during pregnancy, but also early during the postpartum period could help to identify mothers, fathers, and toddlers at risk for psychopathological symptoms.
Line 90: The toddlers had 32-months old…
Line 99: were willing to participate provided a written consent form.
- The structure of the methods section needs some reorganisation. The “participants” sub-section should focus on the selection criteria of the participants. The sociodemographic and clinical profile of the participants should be presented in “results” section.
Author Response
Reviewer’s comment 1
Comments to the Author
This is an interesting paper that seeks to elucidate the way interaction among the parents and their depressive symptoms during the transition to parenthood predict emotional and behavioral problem in toddlers. The research work is sound and findings are well presented. However, I have few minor comments regarding the manuscript.
Response to reviewer’s comment 1
Thank you so much for your comments.
Reviewer’s comment 2
- Kindly recheck the manuscript for language and grammatical errors. To enumerate a few
Line 21: Screening couples’ interaction and depressive symptoms early during pregnancy, but also early during the postpartum period could help to identify mothers, fathers, and toddlers at risk for psychopathological symptoms.
Line 90: The toddlers had 32-months old…
Line 99: were willing to participate provided a written consent form.
Response to reviewer’s comment 2
Amended as suggested.
Reviewer’s comment 3
- The structure of the methods section needs some reorganisation. The “participants” sub-section should focus on the selection criteria of the participants. The sociodemographic and clinical profile of the participants should be presented in “results” section.
Response to reviewer’s comment 3
Amended as suggested.
Reviewer 3 Report
This is an interesting article examining the potential development of psychopathology in toddlers. The study would like to propose an explanatory model of toddlers' emotions and behaviours.
There are some points that authors should clarify in order to enhance their work, in my opinion:
1) "internalizing and externalizing problems" should be described when cited for the first time (i.e. page2);
2) the timing chosen for testing hypothesis, should be supported: exactly, why the 1st trimester of pregnancy, 3 and 30 months were considered? Please provide explanations with further biography to that provided in introduction;
3) could authors give reasons of this study encluded in a "larger longitudianl study"? what is this larger study about? These aspects can help reader in understanding this study's aim and methodology;
4) please support the exclusion criterua for couples "having multiple gestations" (page 3);
5) negative and positive interaction should be defined in the articles's text.
At last, a very minor point, please consider to give a reformulation of sentence on line 37, page 1: "which aim the self-organization" can be not clear, so authors can evaluate an English reformulation.
Thank you for your attention.
Author Response
Reviewer’s comment 1
This is an interesting article examining the potential development of psychopathology in toddlers. The study would like to propose an explanatory model of toddlers' emotions and behaviours.
Response to reviewer’s comment 1
Thank you so much for your comments.
Reviewer’s comment 2
There are some points that authors should clarify in order to enhance their work, in my opinion:
1) "internalizing and externalizing problems" should be described when cited for the first time (i.e. page2);
Response to reviewer’s comment 2
As suggested, we described "internalizing and externalizing problems" when cited for the first time.
Reviewer’s comment 3
2) the timing chosen for testing hypothesis, should be supported: exactly, why the 1st trimester of pregnancy, 3 and 30 months were considered? Please provide explanations with further biography to that provided in introduction;
Response to reviewer’s comment 3
As suggested, we provided support for the timing chosen in the procedures section.
Reviewer’s comment 4
3) could authors give reasons of this study encluded in a "larger longitudianl study"? what is this larger study about? These aspects can help reader in understanding this study's aim and methodology;
Response to reviewer’s comment 4
We added this information in the procedures section.
Reviewer’s comment 5
4) please support the exclusion criterua for couples "having multiple gestations" (page 3);
Response to reviewer’s comment 5
As suggested, we provided support for this criteria.
Reviewer’s comment 6
5) negative and positive interaction should be defined in the articles's text.
Response to reviewer’s comment 6
As suggested, negative and positive interactions were defined in the text.
Reviewer’s comment 7
At last, a very minor point, please consider to give a reformulation of sentence on line 37, page 1: "which aim the self-organization" can be not clear, so authors can evaluate an English reformulation.
Response to reviewer’s comment 7
Amended as suggested.